# Origin of the power-law exponent in the landslide frequency-size

## distribution

Ahoura Jafarimanesh<sup>1</sup>, Arnaud Mignan<sup>1,2</sup> and Laurentiu Danciu<sup>2</sup>

1: Institute of Geophysics, ETH Zurich, Zurich, 8092, Switzerland

2: Swiss Seismological Service, ETH Zurich, Zurich, 8092, Switzerland

Correspondence to: Ahoura Jafarimanesh (Jafarimanesh@erdw.ethz.ch)

**Abstract:** Landslide statistics is characterized by a power-law frequency-size distribution (FSD) with power exponent  $\alpha$  centered on 2.2-2.4, independently of the landslide trigger. So far, the origin of the  $\alpha$ -value, critical to probabilistic hazard

- assessment, remains hypothetical. We present a generic landslide cellular automaton (LSgCA) based on the rules of Self Organized Criticality and on the Factor of Safety (FS) concept. We show that it reproduces the power-law FSD for realistic parameter ranges (i.e. cohesion, soil unit weight, soil thickness, angle of friction, slope angle, pore water pressure) with LSgCA simulations yielding  $\alpha = 2.17\pm0.49$ , which is in agreement with  $\alpha = 2.21\pm0.53$  obtained from an updated meta-analysis of the landslide literature. The parameter  $\alpha$  remains stable despite changes in the landslide triggering process, with
- the trigger only influencing the spatial extent of the landslide initiation phase defined from an FS contour. Furthermore, different FS formulations do not significantly alter the results. We find that  $\alpha$  is constrained during the initiation phase of the landslide by the fractal properties of the topography, as we observed a positive correlation between fractal dimension and  $\alpha$  while  $\alpha$  did not change during the propagation phase of the LSgCA. Our results thus suggest that  $\alpha$  can be estimated directly from the FS map for probabilistic landslide hazard assessment. However full modeling (including the propagation phase)
- would be required to combine the spatial distributions of landslide and exposure in probabilistic risk analysis.

### 1. Introduction

Landslides are a secondary natural hazard (Lee and Jones, 2004; Petley, 2012) that is potentially triggered by weathering (Fuhrmann et al., 2008; Dykes, 2002), rainstorms (Salciarini et al., 2008, Lin et al., 2011), earthquakes (Guzzetti et al., 2005;
Corominas and Moya, 2008; Mignan et al., 2016), volcanic eruptions (Siebert, 1984; Iverson, 1995), or human activities (Ives and Messerli, 1989; Van Den Eeckhaut et al., 2007). The displaced masses due to landslide are categorized as falling, toppling, sliding, spreading, flowing, or by their combination (Sidle and Ochiai, 2006). Landslides represent a dynamic system based on local interactions between the physical characteristics of slope, ground shaking parameters (earthquake case), and/or ground water saturation (e.g., rainfall case).

The landslide statistics is best described by the frequency-size distribution (FSD), which, despite the variety of triggers, appears to systematically follow a power-law probability density function of the form

$$p(x) = (\alpha - 1)x_{\min}^{\alpha - 1}x^{-\alpha}$$

(1)

- with α the power exponent, valid for x ≥ x<sub>min</sub>. Examples based on published landslide inventories are shown in Fig. 1. The
  rollover behaviour observed below x<sub>min</sub> can be described by a double Pareto distribution (Stark and Hovius, 2001) or by the inverse Gamma distribution (Malamud et al., 2004). The different behaviour below x<sub>min</sub> is likely due to a combination of data incompleteness (Stark and Hovius, 2001), as extensively studied in the earthquake case (Mignan, 2012; Kijko and Smit, 2017), and of physical changes (Turcotte et al., 2002; Malamud et al., 2004; Van Den Eeckhaut et al., 2007; Guthrie et al., 2008). However all those models provide similar estimates of α, as illustrated in Fig. 1. As such, we only consider Eq. (1) in
- the present study. While Malamud et al., (2004) suggested a universal value  $\alpha = 2.4$  based on three inventories, a review of about thirty studies found considerable variations with  $\alpha = 2.3\pm0.6$  (Van Den Eeckhaut et al., 2007). Cases of rock fall inventories exhibit different frequency-size distributions (i.e.  $\alpha=1.1$ ) with no rollover effect observed (Malamud et al., 2004). Those cases are not considered in the present study.
  - Of the sheer number of strategies to model landslides (e.g., Densmore et al., 1998; Hergarten and Neugebauer, 1998),
- cellular automata (CA) that follow some of the rules of Self-Organized Criticality (SOC) provide a simple and natural approach to generate power-laws. Indeed, SOC, epitomised by the Bak-Tang-Wiesenfeld CA (Bak et al., 1987) and generalized by the Olami-Feder-Christensen CA (Olami et al., 1992), is characterized by the emergence of a power-law FSD due to a simple bottom-up triggering process analogue to an avalanche. Although SOC CAs and other CA variants are broadly used for landslide modelling (e.g., D'Ambrosio et al., 2006), the sandpile model (Bak et al., 1987), with slope  $\alpha$  =
- 1.0-1.2 does not explain the steeper slope observed for landslides (Hergarten and Neugebauer, 2000; Turcotte et al., 2002). Pelletier et al., (1997) retrieved  $\alpha = 2.6\pm0.1$  from a percolation model controlled by a threshold shear stress dependent on a slope based on physical parameters, but only considered the area of the landslide initiation phase, not of the landslide itself. Interestingly, Hergarten and Neugebauer (2000) obtained  $\alpha = 2.1$  by applying a two-variable product in a SOC system with one relaxation variable and one time-dependent weakening variable, which was related to the factor of safety as an ad-hoc
- assumption in Hergarten (2013). Piegari et al., (2006; 2009) only obtained reasonable  $\alpha$  estimates by allowing arbitrary parameter variations to match the data (see discussion in Hergarten, 2013). It goes the same with Guthrie et al., (2008), who calibrated their input parameters to fit an observed landslide FSD. Those SOC models (Hergarten and Neugebauer, 2000; Piegari et al., 2006; 2009), but also other models (e.g., Densmore et al., 1998), neglected changes in topography by considering only one individual slope, questioning their validity for landslide inventories whose events take place on
- complex topography. Moreover, none of those models lead to landslides that seem very realistic (Hergarten, 2013). Hergarten (2012) proposed a simple model inspired from the basic models of SOC where avalanches propagate on the surface when a certain threshold is exceeded (Bak et al., 1987; Olami et al., 1992) and applied it on real topographies. It was however limited to rock falls and did not define any physical trigger ("random impacts" were used instead). So far it remains

unclear why the landslide FSD seems to be independent of the triggering mechanism and what process or specific physical parameter explains  $\alpha = 2.3\pm0.6$ . Finally, the available models remain mostly abstract and difficult to implement in probabilistic multi-hazard assessment (as in, e.g., Mignan et al., 2014; Liu et al., 2015).

- The aim of this article is to present a generic landslide cellular automaton (in short, LSgCA) that is consistent with the broad range of α-values observed in Nature and is realistic enough for inclusion in future generic multi-risk (GenMR) analyses (e.g. Mignan et al., 2014; 2017; 2018; Komendantova et al., 2014; Liu et al., 2015; Matos et al., 2015). The purpose of the LSgCA is not to replace more sophisticated existing landslide modelling tools but to provide a transparent and simple, yet robust, approach to understand what parameter has the greatest influence on α. This is of importance in the probabilistic hazard assessment of landslides to be able to extrapolate the size of larger potentially damaging events (e.g., Guzzetti et al.,
- 10 2005). We first present the LSgCA approach, where the initiation phase of landslides, the propagation phase and topography simulation are described (section 2). We then update the meta-analysis originally made by Van Den Eeckhaut et al., (2007) by adding about 40 new cases, hence more than doubling the number of data points available to estimate  $\alpha$  (section 3.1). Finally we investigate at what stage of the LSgCA process and for what parameterizations the  $\alpha$  distribution observed in Nature is retrieved (section 3.2).

### 15 2. Landslide generic Cellular Automaton (LSgCA)

#### 2.1. Initiation phase (slope failure)

The initiation phase of landslides can be conveniently quantified via the static factor of safety (FS) (e.g., Crosta, 1998 and references therein). FS  $\leq$  1 represents an unstable slope and can be used for both rainstorm (e.g., Crosta, 1998; Iverson,

- 20 2000) and earthquake triggers (e.g., Newmark, 1965; Jibson, 2007). We test different FS formulations (Table 1) to investigate the role of different conditions of the slope instability in the following states: dry infinite slope, submerged infinite slope, infinite slope with seepage parallel to slope and infinite slope with seepage and tree roots (Cruikshank, 2002; Lambe and Whitman, 1969; Turner and Schuster, 1996; Budhu, 2000; Abramson et al., 1995). The following parameters are involved in the quantification of FS: *C*, cohesion of dry soil;  $\overline{C}$ , cohesion of saturated soil; *t*, thickness of the slide; *d*, vertical
- 25 distance of the sliding body;  $\gamma$ , unit weight of dry soil;  $\theta$ , slope gradient;  $\phi$ , angle of internal friction of dry soil;  $\gamma_t$ , total unit weight material;  $\gamma_{w}$ , unit weight of water;  $\overline{\phi}$ , angle of internal friction of saturated soil,  $\frac{1}{A}\sum_{i=1}^{n} F_i$ , tree roots coefficient. The range of typical material values of the parameters involved in the FS analysis is extracted from the table of soil characteristics by Hall et al., (1994) (Table 2).

An FS map can then be produced that depends mainly on the spatial distribution of  $\theta$ , i.e., on the topography. The impact of 30 earthquake shaking can be modelled via the concept of Newmark displacement (Newmark, 1965):

$$\log D_N = -2.710 + \log_{10} \left[ \left( 1 - \frac{a_c}{a_{max}} \right)^{2.335} \left( \frac{a_c}{a_{max}} \right)^{-1.478} \right] + 0.424M$$
(2)

function of the critical acceleration ratio  $a_c/a_{max}$  and earthquake magnitude *M*, with critical acceleration  $a_c = (FS - 1)g\sin\theta$ and  $g = 9.81 \text{ m/s}^2$  (Jibson, 2007). Landslides are initiated at locations that exceed  $D_N = 15$  cm (Jibson et al., 2000). The earthquake peak ground acceleration  $a_{max}$  is derived from empirical relationships, here using

$$\log_{10}(a_{max}) = b_1 + b_2 M + b_3 M^2 + (b_4 + b_5 M) \log_{10} \sqrt{R_{jb}^2 + b_6^2} + b_7 S_S + b_8 S_A + b_9 F_N + b_{10} F_R$$
(3)

with  $b_1 = 1.43525$ ,  $b_2 = 0.74866$ ,  $b_3 = -0.0652$ ,  $b_4 = -2.7295$ ,  $b_5 = 0.25139$ ,  $b_6 = 7.74959$ ,  $b_7 = 0.0832$ ,  $b_8 = 0.00766$ ,  $b_9 = -0.05823$ ,  $b_{10} = 0.07087$ ,  $S_S = 0$ ,  $S_A = 1$ ,  $F_N = 0$  and  $F_R = 1$  (Akkar and Bommer, 2010).

As investigated in section 3, we do not expect the use of different FS models, nor different triggers, to have an impact on the landslide FSD  $\alpha$ -value since they only impact the spatial extent of the landslide initiation zones defined by the thresholds FS  $\leq 1$  or  $D_N \geq 15$  cm. Indeed, a power law behaviour can emerge whether at the initiation phase by a percolation process on a

10 fractal lattice (e.g., Pelletier et al., 1997) or during the propagation phase in a SOC-style process (see review by Mignan (2011) for the case of earthquake power laws). In the first case,  $\alpha$  depends on the fractal dimension of the lattice, in the second, on the propagation rules that depend on soil conditions, and not on the triggering mechanism, at least not at the lowest-order.

#### 2.2. Propagation phase (post-failure motion)

#### 15

The proposed CA is based on the concept of sandpile (Bak et al., 1987) and defined for a square grid composed of cells (x,y) binned in  $\Delta s$  increments. Variables are the altitude z(x,y) and soil depth h(x,y). Input parameters are the initial topography (z, h) and soil conditions ( $\phi$ , *C*,  $\gamma$ , etc.) from which FS is computed (Tables 1-2). We use the Moore neighbourhood nomenclature (Gray and New, 2003), and compute the maximum slope  $\theta_{max}$  in the direction  $dir_{Moore}(\theta_{max})$ . In contrast to the SOC random field where triggering occurs at random cells, triggering is here initiated in cells for which conditions  $FS \leq 1$  or  $D_N \geq 15$  cm are satisfied (section 2.1). The landslide footprint is defined as  $LS_{RS}(x,y \mid FS \leq 1) = 1$  for rainstorm triggers and as  $LS_{RS}(x,y \mid D_N \leq 15 \text{ cm}) = 1$  for earthquake triggers, and with LS(x,y) = 0 elsewhere.

For LS(x,y) = 1 and h(x,y) > 0, the propagation rule is

$$\begin{cases} z(x,y) = z(x,y) - \Delta h \\ h(x,y) = h(x,y) - \Delta h \\ z(dir_{Moore}[\theta_{max}(x,y)]) = z(dir_{Moore}[\theta_{max}(x,y)]) + \Delta h \\ h(dir_{Moore}[\theta_{max}(x,y)]) = h(dir_{Moore}[\theta_{max}(x,y)]) + \Delta h \end{cases}$$
(4)

25 where the mass movement is defined by

 $\Delta h = [z(x, y) - z(dir_{Moore}[\theta_{max}(x, y)]) - \Delta s \tan(\theta_{stable})]/2$ (5)

The mass is progressively transferred from cells to cell by  $\Delta h$  such that a stable slope  $\theta_{stable}$ , if *h* is high enough, is reached. If *h* is too low, i.e.,  $h 

5

unstable cells reaches a plateau (i.e., stability criterion to avoid infinite loops). It should be noted that the process is non-SOC since it naturally dies off after any event. However, the constant load defined from successive earthquakes and/or rainstorms might lead to a SOC-like behaviour on the long-term (Hergarten, 2003).

Figure 2 shows a landslide generated by the LSgCA in a smooth virtual region, as defined in Komendantova et al., (2014), Liu et al., (2015) and Mignan et al., (2017) for generic multi-hazard testing, here with an earthquake as trigger (black segment). Note that the propagation phase, in itself, does not lead to a power-law distribution (nor does the initiation phase

- for the matter, as explained in the previous section). In contrast to SOC where the spatial field is random and the load increased by random increments, the proposed CA can apply on smooth surfaces and produce relatively realistic landslide footprints. Loading, by an increase or addition of an elliptic instability patch (for the earthquake trigger), may yield only one
  landslide, as shown in Fig. 2 (see video in the supplementary materials).
- For the rest of this study, we will apply the LSgCA on more realistic topographies and investigate whether the  $\alpha$  distribution observed in Nature is explained from the topography itself (i.e., percolation model) and whether it is altered by the landslide propagation phase (i.e., SOC-like model). We model the fractal topography using the diamond-square algorithm (Fournier et al., 1982), based on fractional Brownian motion with  $\sigma_n = 2^{-nH}\delta$ ,  $\delta$  being the *z*-deviation at first iteration, *n* the iteration
- 15 number, and *H* the Hurst exponent. The fractal dimension is simply  $D_f = 3$ -*H*. We will test  $2.1 \le D_f \le 2.9$ , representing increased terrain roughness with 7 iterations leading to a  $2^7+1 = 129 \times 129$  cells grid with the constant thickness *h*. The resulting topography is considered "pristine" and is unrealistic due to its many steep peaks (Miller, 1986) and to the constant *h*. However, we can then apply the LSgCA to "erode" the topography, which yields a new topography that is more realistic (i.e., scarps become devoid of soil while basins contain more soil). We define the topography on a 10 by 10-km grid with a
- 20 10-meter resolution. Individual landslides are defined as continuous patches verifying LS(x,y) = 1. The  $\alpha$  parameter is then calculated using Maximum Likelihood Estimation (MLE), as described in Clauset et al., (2009).

### 3. Results

#### 3.1 Updated meta-analysis

- To validate our model based on FSD statistics (section 3.2), we first update the review made by Van Den Eeckhaut et al., (2007). We do so by adding observations made during the last decade. Inputs are listed in Table 3 and the resulting distribution plotted in Fig. 3. This new meta-analysis includes a total of 70 cases, including 40 new compared to the 2007 study. Note that  $\alpha = \alpha_{cum}$ +1 when the FSD is cumulative but also that  $\alpha = 1.5\alpha_V$  when the landslide size is volume V instead of area A (with  $V \sim A^{3/2}$ ). When compared to other models,  $\alpha = \alpha_2$ +1 =  $\rho$ +1 with  $\alpha_2$  from the double Pareto distribution
- 30 (Stark and Hovious, 2001) and  $\rho$  from the inverse Gamma distribution (Malamud et al., 2004).

We finally obtain the distribution  $\alpha = 2.21\pm0.53$ , in very close agreement with the original result  $\alpha = 2.3\pm0.6$  of Van Den Eeckhaut et al., (2007). The p+1 values obtained for the three landslide inventories shown in Fig. 1 (Harp and Jibson, 1995; Xu et al., 2015; Bucknam et al., 2001) are also plotted.

## **3.2. LSgCA application on fractal topography**

5

We ran simulations in which the ground parameters were drawn randomly from the range of values presented in Table 2. The four FS formulations of Table 1 were systematically tested for sensitivity analysis. We first evaluated  $\alpha$  at different steps of the landslide process (with constant  $D_f = 2.4$ ): (1) at the initiation phase, directly based on the FS map, assuming that the landslide area limits itself to FS 

5

(as identifiable from the FS maps in Fig. 4). In a second test, we evaluated the role of  $D_f$  changes (in the range of  $2.1 \le D_f \le 2.9$ ) on the estimation of  $\alpha$  in the percolation phase (i.e., initiation phase, no propagation). The higher values of  $D_f$  in the percolation phase imply an increase in the surface roughness. This suggests that we limit the range of our analysis to  $2.2 \le D_f \le 2.5$  to produce more realistic topographies in order to interpret the behaviour of  $\alpha$  (lower and higher values represent either too smooth or too rough topographies – see below). Similar to the analysis in the previous sections, the ground parameters were chosen randomly from the typical range of parameters in Table 2. Once again, the four FS formulations were tested (Table 1). Results are shown in Fig. 6. The boxplot indicates the range, the lower quartile, the median, and the upper quartile of the produced  $\alpha$  obtained from a total of 36,000 simulation (1,000 per FS, per  $D_f$ ). The surface topography of the case  $D_f = 2.1$  is too smooth to trigger landslide and produce a meaningful  $\alpha$  distribution. The median of the  $\alpha$  values observed in the

10 range  $2.2 \le D_f \le 2.5$  tend to increase for all FS equations. For the case of  $D_f = 2.6$ , the median value of the  $\alpha$  drops down to an approximate value of  $\alpha = 1.5$ , due to the high instability of fractal topography and therefore the failure of an individual mega large landslide. A similar result is obtained for  $D_f = 2.7$ . Above, the fractal topography fails to produce a realistic surface where landslides analysis is able to produce valid results. In the acceptable range of the fractal dimension ( $2.2 \le D_f \le$ 2.5), a higher  $D_f$  means an increase of the ratio of small landslides to the larger ones until they coalesce onto the full

15 topography grid, explaining our observations.

#### 4. Conclusions

In this study, we presented a landslide generic cellular automaton (LSgCA) that models the propagation phase of landslides. After revisiting the literature, we updated the observed power-law exponent distribution to  $\alpha = 2.21\pm0.53$  and validated our

- LSgCA by obtaining a similar distribution in simulations with  $\alpha = 2.17\pm0.49$ . Although realistic values had already been obtained in previously published models (e.g., Pelletier et al., 1997; Hergarten and Neugebauer, 2000), we here demonstrated that  $\alpha$  remains at first-order constant at the different stages of the landslide process. Moreover, despite the  $\alpha$  variability being due to different input parameter values and different topography iterations, an increasing trend in the median  $\alpha$  was observed as a function of increasing fractal dimension  $D_{f_0}$  suggesting this parameter as the main parameter
- influencing  $\alpha$ . Indeed, since  $\alpha \approx 2.2$  is already observed in the initiation phase where the only information is the FS map,  $\alpha$  can only find its origin in the underlying fractal topography. As a corollary, it also explains why the landslide trigger type (e.g., weathering, rainfall, earthquake) does not seem to influence  $\alpha$ .

We believe that the LSgCA solves the existing problem of abstract physical-based scenario analyses, unrealistic topographies and/or the lack of landslide propagation phase in the landslide CA literature (Piegari et al., 2006; Hergarten and

30 Neugebauer, 1998; Densmore et al., 1998; Guthrie et al., 2008). LSgCA provides a simple tool for further studies in probabilistic multi-hazard assessments by defining landslide footprints that could be included in stochastic landslide sets conditional on rainstorm or earthquake triggers (e.g., Mignan et al., 2014; 2018; Liu et al., 2015; Matos et al., 2015). But by

proving that the origin of  $\alpha$  lies in the topography, landslide hazard can be approximated directly from the FS maps, before any dynamic modelling.

## 5. Acknowledgment

5 This work was supported by the Swiss Competence Center for Supply of Electricity (SCCER SoE) T4.1 "Risk, safety and societal acceptance" and by the Swiss National Science Foundation Program "NRP70" as a part of the project "Risk Governance of Deep Geothermal and Hydro Energy" with the grant number 40 7040\_153931.

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
