# Peer review of "Origin of the power-law exponent in the landslide frequency-size"

_Natural Hazards and Earth System Sciences, 2018_

## Referee Comment (RC1) · S. Hergarten (Referee) · 9 Jul 2018

The manuscript addresses the scaling exponent of the power law distribution that has been widely found for landslide sizes. As this power-law distribution appears to be universal, understanding its origin could indeed be a major steps towards understand landslide dynamics and finally improve hazard assessment. So the topic is clearly important and fits very well into the scope of NHESS.

As the topic is also close to by personal interests I enjoyed reading the paper. However, although I am familiar with the topic I had difficulties at many points, so that I am afraid that the majority of the readers will not be able to follow. My feeling is that the explanations are much too short.

[Figure]

Beyond this, I am not completely convinced whether the results are really as good as they look, and a more thorough discussion is needed. The main result seems to be that the fractal topography makes the power-law distribution and the propagation process has a minor effect. In this context I would request the following questions to be answered.

1. Several approaches for calculating the factor of safety (FS) are considered. However, as long as the parameters are constant, FS is a function of the local slope. What would be the result if we just assume that points where the slope exceeds a given threshold are unstable instead of the rather complicated FS?

2. If the result was basically the same if we only measured the sized of patches steeper than a given threshold slope angle as suggested in (1), would the result be surprising for an artificial fractal topography? And how close would a real topography come to this?

3. The thickness of the soil layer is rather low at least in the pristine fractal topography. This means that even removing the whole soil at one point cannot lead to a very strong decrease of the FS at the upward neighbors for the 10 m grid spacing used here. So is it really surprising that the propagation does not have a significant influence compared to the initiation? And would this change if we used a finer grid at the same soil thickness?

4. The statistics seem to be rather small, and the range of sizes where the power law distribution is visible is rather narrow. Why are the data sets so small?

In the following I list some more points where the explanation should be improved, provided that the points raised above can be addressed.

**Page 1, line 23:** I would not consider weathering as a trigger, but rather as a long-term driving force.

**Page 2, lines 21–22:** The results of Pelletier et al. (1997) should be explained a bit more in detail in order to see more clearly in which sense the new results differ.

**Page 5, line 1:** I did not get what it means that the number of unstable cells reaches a plateau.

**Page 5, line 28:** I thought a relationship like $\alpha = 1.5\alpha_V$ would be valid in the cumulative sense, which would imply $\alpha - 1 = 1.5(\alpha_V - 1)$ instead of the relationship given in the manuscript.

**Table 1:** It would be good to include a short list of the parameters here.

**Figure 1:** The three plots are so narrow that it is not easy to read them.

**Figure 2:** What is this topography good for in the context of this manuscript?

**Figure 4:** The meaning of $m$ is not immediately clear here.

However, I would like to emphasize that these points are only details, while the four aspects mentioned above are more fundamental for me.

---

## Author Comment (AC1) · 24 Aug 2018

Dear Professor Stefan Hergarten,

Thank you for your comments on the discussion paper by Jafarimanesh et al. (2018). Below is our three-part answer: (1) to clarify the role of the local terrain slope relative to the more complex factor of safety (FS) in the emergence of the power-law, (2) to show the role of soil thickness on the simulated frequency-size distribution, and (3) to respond to all other comments and questions.

1 Factor of safety versus constant slope threshold

**1) FS is indeed a function of the local slope $\theta$ if all other soil parameters are kept constant. We thus now test $20° \leq \theta \leq 40°$ as a possible interval of unstable slope thresh-**

[Figure]

olds, based on the 4 FS formulations and all other parameters kept within reasonable ranges. Figure A.1 shows the frequency-size distributions derived from the size of unstable areas, defined by $\theta \geq 20°$ (left) and $\theta \geq 40°$ (right). We see that the power-law behaviour is still observed (with median exponent $\alpha$ = 1.9-2.2 within the range of fluctuations observed for the percolation model based on FS). The impact of increasing the $\theta$ threshold is mainly a decrease in the overall number of landslides, since there is less soil (smaller area) made unstable in the process. Therefore, our conclusion holds; the power-law behaviour must emerge from the fractal topography since the $\theta$ spatial distribution is a direct consequence of it. However, it is preferable to consider the FS threshold in landslide modelling since it takes into account the other soil characteristics (refined model in which the $\theta$ threshold can change locally) and since FS is a standard in landslide hazard assessment. #2) This result is consistent with a percolation model in which a change of the threshold would only change the number of clusters observed and not their size distribution since a fractal is self-similar. We expect a similar result for real topography, which is also fractal (e.g., Mandelbrot, 1983). We did the test with some real topographies in Switzerland and found a similar power-law behaviour. The power-law would however disappear if the topography was smooth (see e.g., our original figure 2 where only one landslide footprint emerges). In the revised manuscript, we will therefore emphasize that the power-law already emerges if we use directly a $\theta$ threshold instead of FS but that in practice, FS should still be used as triggering value.

**2 The role of soil thickness**

**3) We originally used a constant h = 10 m in the pristine fractal topography so that the final soil depth distribution is realistic with 0 < h < 100 m. This distribution is shown in Figure A.2. We now additionally tested an initial h = 25 m, with results shown in Figure A.3. We observe a decrease in the power-law exponent $\alpha$, controlled by more events in the tail. The same exercise on an FS-based percolation model shows only a general increase in the number of landslides and no change in $\alpha$. This suggests that the propagation phase has indeed an impact on $\alpha$, as suggested by the reviewer. We will**

therefore update our abstract and conclusion to mention that while the power-law be-haviour indeed emerges from the topography via the initiation phase, the total amount of soil available in any given region will tend to decrease $\alpha$ by filling the distribution tail (leading to more greater landslides) via the propagation phase. We thank the reviewer for spotting the role of the initial h. It also means that the FS map can still be used as a proxy to landslide hazard but with the resulting $\alpha$ only giving an upper bound to the true $\alpha$. We will add a new figure showing the respective role of h for different fractal dimensions.

**4) The statistics obtained in our study is observed over about two orders of magnitude (our original figure 4), which is similar to the one observed in Nature (our original figure 1). We believe that the concern of a rather narrow distribution and of small landslide sizes is due to a typo from our part. The unit of the x-axis is not "squared metre" but "number of cells" in the model. In our original figure 4, based on a grid of 78m resolution, we get 50 cells = 304,200 m2 (which makes our maximum median estimate consistent with the real observations of the figure 1). We apologize for this error and will correct it in the revised version.**

3 Response to the other comments

p1, l23: We will change "weathering as a trigger" to "weathering as a long-term driving force" in the revised manuscript.

p2, l21-22: We will now describe the work of Pelletier et al. (1997) as follows (same as original in italics): "Pelletier et al. (1997) retrieved $\alpha$ from a percolation model controlled by a threshold shear stress dependent on the terrain slope and on other physical parameters (such as cohesion, internal friction angle, etc.), which is therefore similar to the FS threshold approach. However, the origin of the power-law is difficult to assess there since the authors combined two fractal processes, the topography (to compute the slope angle) and the soil moisture (to compute the soil parameters). That study also only considered the area of the landslide initiation phase, not the landslide

itself (i.e., no propagation phase)." Our results are roughly similar to Pelletier et al. (1997), as already noted p6 (lines 9-10, 15-16), but by including the landslide SOC-like process, we were able to clarify the origin of $\alpha$, which is the main novelty of our work. Also, while both used a fractal topography, we used a random uniform distribution of soil parameters instead of a fractal one. This will be clarified.

p5, l1: It is possible that a few cells remain unstable with a soil element jumping between cells in an infinite loop. Therefore, we fixed a break defined as the iteration at which the number of unstable cells has remained constant over the previous 3 iterations (this number is close to zero and is often equal to zero, i.e., full grid stable after a relatively small number of iterations). This will be clarified in the revised manuscript.

p5, l28: We will correct $\alpha$ to $\alpha$cum as we indeed have $\alpha$cum = 1.5 $\alpha$v - which is what we used before applying $\alpha$non-cum=$\alpha$cum+1, as in Guzzetti et al. (2002).

We will provide the FS parameters in Table 1 of the revised version.

Figure 1: We will reformat the three plots to stretch them in the x-direction

Figure 2: This figure illustrates that the propagation of a landslide, which follows a SOC-like behaviour, does not necessarily leads to a power-law behaviour (since only one landslide footprint is created). This is already explained p5, l6-10. This allows to decouple the roles of the fractal topography and of the cellular automaton. This will be clarified in the text.

Figure 4: The parameter 'm' was originally used in the FS equation in an earlier version of the manuscript as the percentage of a slope that is saturated. In the latest version, we updated the FS equations and therefore removed the explanation about 'm' from the original manuscript. We will update Figure 4 in the revised version of the manuscript to remove this legacy parameter.

References:

Guzzetti, F., Malamud, B. D., Turcotte, D. L., and Reichenbach, P.: Power-law correlations of landslide areas in Central Italy, Earth Planet. Sci. Lett., 195, 169–183, doi: 10.1016/S0012-821X(01)00589-1, 2002 Jafarimanesh, A., Mignan, A., and Danciu, L.: Origin of the power-law exponent in the landslide frequency-size distribution, Nat. Hazards Earth Syst. Sci. Discuss., doi: 10.5194/nhess-2018-167, 2018 Mandelbrot, B. B.: The Fractal Geometry of Nature, Macmillan, New York, 1983

Figures

Figure A.1. Role of different unstable slope $\theta$ thresholds on the landslide frequency-size distribution. The power-law behaviour still emerges from the $\theta$ spatial distribution, which depends directly on the fractal topography.

Figure A.2. Distribution of soil thickness h after the erosion step of the LSgCA, applied to a fractal topography with initial constant h = 10 m.

Figure A.3. Role of total soil available represented by different initial soil thicknesses h = 10 or 25 m. More material yields more larger landslides via the LSgCA propagation phase, not observed for h = 10 m (nor by the initiation phase for h = 25 m).

[Figure]

[Figure]

**Fig. 1.**

[Figure]

**Fig. 2.**

[Figure]

**Fig. 3.**

---

## Referee Comment (RC2) · Anonymous Referee #2 · 18 Oct 2018

This paper aims to provide a physical understanding regarding the factors controlling power-law exponent, which is widely observed in frequency-size distributions of landslides. Although the power-law relation is observed in a number of landslide inventories, the reasoning behind this relation and the factors controlling power-law exponent are still under discussion. Therefore, this paper addresses an interesting scientific question, and the paper has a fit to NHESS. I think this study could possibly be publishable with major revisions.

General comments:

1. The paper is missing some key literature that aims to provide a physical explanation for the power-law distribution of landslides. For example, Liucci et al. (2017) (http://dx.doi.org/10.1016/j.geomorph.2017.04.017) introduced a model: "the model is

capable of reproducing the scaling behavior of real landslide areas and suggest that topography is a good candidate to explain their scale-invariance." On the other hand, Stark and Guzzetti (2009) (doi:10.1029/2008JF001008) and Frattini and Crosta (2013) (http://dx.doi.org/10.1016/j.epsl.2012.10.029 ) use geotechnical parameters to explain the power-law exponent of the landslide size distribution. For example, Frattini and Crosta (2013) argue that "the exponent of the power-law tail is controlled by both the topography and the depth profile of material strength." I think the authors should elaborate on these papers and indicate the points that they agree with these papers. The authors then need to indicate what is missing in the literature; what they argue differently than the previous studies.

2. The authors also need to indicate why exploring the factors controlling power-law exponent is important. There is one line in the abstract saying that "So far, the origin of the $\alpha$-value, critical to probabilistic hazard assessment, remains hypothetical." This may be enough for the abstract. However, in the introduction, they should explain it further because there is only one line at the end of the introduction saying that "This is of importance in the probabilistic hazard assessment of landslides to be able to extrapolate the size of larger potentially damaging events."

3. I am aware of the large literature examining the power-law relation that assumed to be valid for landslide frequency-size distribution. However, I think the authors should note that this is still an assumption. For example, in page 2, line 2, the authors indicated that "the frequency-size distribution (FSD), which, despite the variety of triggers, appears to systematically follow a power-law probability density function." I believe it is still up for debate whether that is, in fact, the case, especially if there is a physical mechanism responsible for deviation from power-law scaling. Therefore, it might be better to emphasize that this is what we observe most of the time. However, for example, Tanyas et al. (2018) (DOI: 10.1002/esp.4359) examine the validity of power-law relation using KS-test and show that "...six out of the 45 inventories have P-values lower than 0.1", so the power-law fit may not be a plausible hypothesis (?). Thus, the

authors need to mention that it is not a fact, but a general observation.

4. The authors have a nice literature summary showing power-law exponents reported in the literature. However, there are still some missing papers. For example, ten Brink et al. (2006) (doi:10.1029/2006GL026125) analyzed sub-marine landslides. Tanyaş et al. (2018) (DOI: 10.1002/esp.4359) examined 45 landslide inventories and reported that power-law exponents of earthquake-triggered landslide inventories range from 1.8 to 3.7.

5. I would like to be informed by the limitations of the model. This is a simplified model to understand a natural process. So, there must be some limitations, and I think the authors should list them in the manuscript. For example, in Table 2, some geotechnical parameters are presented and used to calculate FS in the modeling stage. This is quite a simplification, and I do not mean that the author can not use them but if you use them, then its limitations should be addressed as well. In this regard, somewhere before the Conclusion section, the authors should consider adding a Discussion section to discuss these issues. The things I mentioned in the previous item (4) can also be discussed in a Discussion section.

6. There are some basic concepts mentioned in the manuscript without giving enough background information. For example, what is Df? What is the Hurst exponent? Beyond its role in equation (1) what power-law exponent stands for? What is the physical meaning of having larger or smaller beta or Df, etc.? Similarly, what do you mean by landslide propagation phase? I think the authors should give some background about such terms in the introduction section, so the reader can understand why you are telling about, for example, propagation phase, and what it refers to?

7. In terms of the structure, I think it would be better the paper has a clear methodology, results and discussion sections. In the current version the discussion section is missing, and I think it is a must and you should add a discussion section (please see my comments below). Also, in the result section, there are some parts that need

to be moved to the method section. For example, in page 5, line 25, authors say that "To validate our model based on FSD statistics (section 3.2), we first update the review made by Van Den Eeckhaut et al. (2007)." Similarly, in page 6, line 7, the authors say that "We first evaluated $\alpha$ at different steps of the landslide process: (i)....(ii)." These are just examples for the parts that need to be moved to the methodology section.

8. The authors used the mean values to validate their approach: (In page 7, line 19) "After revisiting the literature, we updated the observed power-law exponent distribution to $\alpha$ = 2.21±0.53 and validated our 20 LSgCA by obtaining a similar distribution in simulations with $\alpha$ = 2.17±0.49." Also, for example, in figure 5 authors show that the range of estimated and real exponents match. The question in my mind then why do not the authors select one or two landslides affected area that authors have landslide inventories to estimate those particular cases. For example, the authors can examine the sites of which landslide frequency-size distributions are presented in Figure 1. Alternatively, some other sites that the authors have access to their environmental conditions (e.g., lithology, hydrology, etc.) to estimate the parameters need to run the FS equation. Then the authors would make an event-based validation. This is just an idea, and I am wondering it is possible or not.

Line by line comments:

Page 2, Line 9. "However all those models provide similar estimates of $\alpha$, as illustrated in Fig. 1". I do not agree with this statement; please take a look at my general comment (4). There are cases that we can observe different power-law exponents.

Page 2, Line 23. "but only considered the area of the landslide initiation phase, not of the landslide itself." Please elaborate further, not so clear what you mean by this sentence. What do you mean by "area of the landslide initiation phase" and "landslide itself."

Page 2, Line 25. "Piegari et al., (2006; 2009) only obtained reasonable $\alpha$ estimates by allowing arbitrary parameter variations to match the data (see discussion in Hergarten,

2013)" What for you mean by "reasonable"? The typical range of previously observed values for landslide inventories (1.4-3.4), which have a central tendency of around 2.3–2.5 (Van Den Eeckhaut et al., 2007; Stark and Guzzetti, 2009). So, we have power-law exponents range from 1.4-3.4, then what is reasonable. It might be better if you give a further explanation instead of directing us to the discussion section of Hergarten, 2013.

Page 3, Line 29. "An FS map can then be produced that depends mainly on the spatial distribution of $\theta$, i.e., on the topography." This is not clear to me; please explain a bit. You are presenting us FS equations where FS depends on many variables, and then you are saying that FS mainly depends on $\theta$. Why?

Page 3, Line 4. "As investigated in section 3, we do not expect the use of different FS models, nor different triggers, to have an impact on the landslide FSD $\alpha$-value since they only impact the spatial extent of the landslide initiation zones defined by the thresholds FS $\leq$ 1 or DN $\geq$ 15 cm." This is not clear to me; please elaborate further.

Page 5, Line 23. "$\alpha = 1.5\alpha V$ when the landslide size is volume V instead of area A (with V $\sim$ A3/2)." You need to give a reference here. You can use various scaling exponent in the equation between landslide volume and area (V=$\alpha$AˆÉč). Please give a look at the literature (e.g., Guzzetti et al., 2009; Klar et al., 2011; Larsen et al., 2010) and let us know the reference you used.

Page 6, Line 1. "We finally obtain the distribution $\alpha = 2.21\pm0.53$" Please rephrase your sentence and indicate that this is the average value you calculated from the literature survey and please also refer to Figure 3.

Page 6, Line 2. "We ran simulations in which the ground parameters were drawn randomly from the range of values presented in Table 2." I think Table 2 deserves more explanation. You listed the geotechnical material properties of four different rock types. However, in a slope stability problem, the shear strength parameters are generally controlled by rock mass parameters, not the rock material parameters (e.g., Hoek and Brown, 1980). If the slope failure occurs along a discontinuity surface then the shear

strength parameters of the discontinuity surface are used (e.g., Barton and Choubey, 1977). Please elaborate further why did you use these material properties and considered only these four rock types.

Minor comments:

Please go through your references and drop the "," coming after et al. (like the example I gave below) Page2, line 26. Guthrie et al., (2008)

Figure 1. The x-axis does not refer landslide count; it should be landslide frequency that is count per interval. Moreover, please indicate its unit (mˆ-2)

Figure 2. You do not say any word about liquefaction that you indicated in this figure. I think you may better describe this figure and the supplementary material you provided.

Figure 3. Please make your labels a bit bigger and readable. Moreover, please use the correct symbol for the power-law exponent in the figure.

Figure 5. Please put a label for the y-axis.

Figure 6. Please put a label for the y-axis.

References Barton N, Choubey V. 1977. The shear strength of rock joints in theory and practice. Rock mechanics 10: 1-54. DOI: 10.1007/bf01261801 Guzzetti F, Ardizzone F, Cardinali M, Rossi M, Valigi D. 2009. Landslide volumes and landslide mobilization rates in Umbria, central Italy. Earth and Planetary Science Letters 279: 222-229. DOI: https://doi.org/10.1016/j.epsl.2009.01.005 Hoek E, Brown ET. 1980. Empirical strength criterion for rock masses. Journal of Geotechnical and Geoenvironmental Engineering 106: 1013-1035 Klar A, Aharonov E, Kalderon‐Asael B, Katz O. 2011. Analytical and observational relations between landslide volume and surface area. Journal of Geophysical Research: Earth Surface (2003–2012) 116. DOI: 10.1029/2009JF001604 Larsen IJ, Montgomery DR, Korup O. 2010. Landslide erosion controlled by hillslope material. Nature Geoscience 3: 247-251. DOI: 10.1038/ngeo776

---

## Author Comment (AC2) · 2 Nov 2018

Dear reviewer,

Thank you for your comments on the discussion paper by Jafarimanesh et al. (2018) and for providing a list of recent publications on the same topic. Please find below our answers to your comments:

1) Literature on landslide power-law distribution

We will add the following references: Liucci et al. (2017); Stark and Guzzetti (2009); Frattini and Crosta (2013). We will mention these studies in the Introduction with other landslide models, as well as in the new Discussion section. We note that Liucci et al.'s results agree with ours, i.e., that the topography is key in the origin of the power-law

behaviour, which provides a validation of the two models, and verifies the hypothesis postulated by Frattini and Crosta. Our study remains the first to explore the changes of $\alpha$ per modelling step (topography slope, initiation phase, propagation phase), and to investigate the direct role of the fractal dimension of the topography. Moreover, using the factor of safety as main controlling slope failure metric avoids the use of ad-hoc parameters, as done in some previous CA studies.

2) The importance of the power-law exponent in hazard assessment

We will develop more on the importance of the power-law exponent in the introduction: After "This is of importance in the probabilistic hazard assessment of landslides to be able to extrapolate the size of larger potentially damaging events", we will now add: "Indeed, as already mentioned by Liucci et al. (2017), $\alpha$ provides the mean to estimate the probability of occurrence of landslides of different magnitudes, including landslide sizes greater than experienced in the past. An increase in $\alpha$ means a decrease in the ratio of larger event sizes. This is analogue to the Gutenberg-Richter law in earthquake risk where the power exponent (or b-value in exponential scale) is the most critical parameter with the earthquake rate."

3) On the (non-)universality of the power-law

First, in "the frequency-size distribution (FSD), which, despite the variety of triggers, appears to systematically follow a power law probability density function", we will change "systematically" by "reasonably".

Second, we will clarify that the power-law behaviour is only an approximation of the landslide FSD behaviour and that other functions have been proposed, such as the double Pareto and inverse Gamma. Those functions consider the roll-over at small sizes but can also explain a potential curvature in log-log space at the tail of the FSD - this will be better illustrated in the new version of Figure 1. More generally, while using the power-law provides a proxy to the true distribution, it should be emphasised that the universality of the power-law is now contested in various domains (Broido

and Clauset, 'Scale-free networks are rare', 2018), including landslides. Tanyas et al. (2018) showed on a large landslide inventory that the FSD is much more variable than previously assumed. Only part of this variability can be explained by the $\alpha$ range. We here use the '$\alpha$ proxy' to be consistent with previous studies spanning from 1969 to 2013.

4) Update of the landslide meta-analysis

We will update our reference database with Brink et al. (2006) and Tanyas et al. (2018) and update our results accordingly. The recent result of $1.8 < \alpha < 3.7$ is in agreement with our review of $1.7 < \alpha < 2.8$ (1-sigma range).

5) Limitations of the LSgCA model

The limitations are the following: • Landslides are considered instantaneous since the successive increments in the cascading process are not related to specific time intervals. • In contrast to some other CAs, no time-dependent weakening is used. Instead the decrease in the slope angle stability threshold is quantified by increasing the water saturation parameter in the factor of safety. This parameter is kept constant for any given CA run. • Water saturation is assumed homogeneous in space, illustrative of loading per rainfall but unrealistic locally, ignoring water flow. • Tectonic uplift is neglected, but negligible at the temporal scale of individual landslides. • The soil characteristics are homogeneous along z.

They will now be listed in the new Discussion section.

6) Background information development

More background information will be given for the following terms:

Df is the fractal dimension of the topography, here assumed constant in the studied area. It can be understood as the degree of roughness of the topographic surface, via Df = 3-H where H is the Hurst exponent. Therefore the 2 parameters are anti-correlated with an increase of roughness representing a decreasing H and increasing

Df. The Hurst exponent is a measure of the randomness of the stochastic process with $H = 1/2$ representing the standard Brownian motion.

More details about the power exponent $\alpha$ are given in answer #3.

The landslide propagation phase corresponds to the cellular automaton process of mass transfer between the grid cells. The initiation phase therefore does not include any SOC behaviour. In this case, the power-law can emerge from the spatial distribution of the factor of safety that is function of the fractal topography.

7) Structure of the article

We will rearrange the paper using the common Method/Results/Discussion structure, considering the detailed remarks of the reviewer.

8) Real topography validation

Real topography validation is the subject of another manuscript currently in preparation. We tested the model on the Illhorn slope located in the Illgraben catchment, in Swiss Valais, and obtained results in agreement with observations. Interestingly the lower $\alpha$ observed there is retrieved by the LSgCA by refining the site-specific soil characteristics. The preliminary result is shown in Figure B.1.

Fig. B.1. Simulated landslides with LSgCA on the Illgraben catchment topography. For the quartzite site conditions, we obtained $\alpha = 1.7\text{-}1.9$ in agreement with observations of real landslide inventories (Bennett et al. 2012).

Minor comments: we will update the manuscript accordingly.

[Figure]

**Fig. 1.**